# Effects of a 16-Week Green Exercise Program on Body Composition, Sleep, and Nature Connection in Postmenopausal Women

**DOI:** 10.3390/ijerph22081216

**Published:** 2025-08-01

**Authors:** Helena Moreira, Chiara Tuccella, Emília Alves, Andreia Teixeira, Carlos Moreira, Irene Oliveira, Valerio Bonavolontà, Catarina Abrantes

**Affiliations:** 1Department of Sport Science, Exercise and Health, University of Trás-os-Montes and Alto Douro, 5000-801 Vila Real, Portugal; andreiat@utad.pt (A.T.); carfute@gmail.com (C.M.);; 2Research Center in Sports Sciences, Health Sciences and Human Development (CIDESD), University of Trás-os-Montes and Alto Douro, 5000-801 Vila Real, Portugal; 3Centre for the Research and Technology of Agro-Environmental and Biological Sciences—CITAB, Inov4Agro University of Trás-os-Montes and Alto Douro—UTAD, Quinta de Prados, 5000-801 Vila Real, Portugal; ioliveir@utad.pt; 4Department of Applied Clinical and Biotechnological Sciences, University of L’Aquila, 67100 L’Aquila, Italy; chiara.tuccella1@student.univaq.it (C.T.); valerio.bonavolonta@univaq.it (V.B.); 5Department of Neurosciences, Biomedicine and Movement Sciences, University of Verona, 37131 Verona, Italy; 6Department of Sports Science, Douro Higher Institute of Educational Sciences, 4560-708 Penafiel, Portugal; alvesemil@gmail.com; 7Research Centre in Physical Activity, Health and Leisure (CIAFEL), Faculty of Sport at the University of Porto, 4200-450 Porto, Portugal; 8Department of Mathematics, University of Trás-os-Montes and Alto Douro, 5000-801 Vila Real, Portugal; 9Center for Computational and Stochastic Mathematics, CEMAT-IST-UL, University of Lisbon, 1600-214 Lisbon, Portugal

**Keywords:** menopause, nature relatedness, sleep, adiposity, muscle condition

## Abstract

Physical activity, particularly when practiced in natural settings, has well-established benefits for overall health, sleep, and body composition. These effects are especially important for postmenopausal women, although research specifically targeting this population remains limited. The study evaluated a 16-week multicomponent outdoor exercise program (cardiorespiratory, strength, balance, coordination, and flexibility training) in postmenopausal women, consisting of three 60 min sessions per week. Participants were non-randomly assigned to an experimental group (EG, n = 55) and a control group (CG, n = 20). Measurements were taken at baseline and after 16 weeks, including body composition, sleep (duration and quality), and connection with nature. No significant differences were observed between groups at baseline. After the intervention, the EG and CG presented significant differences (*p* ≤ 0.01) in the rates of change in body mass, fat mass (FM; −9.26% and −1.21%, respectively), and visceral fat level (VFL; −13.46 points and −3.80 points). These differences were also observed for the sleep fragmentation index (*p* ≤ 0.01), but not for connection with nature. A significant interaction effect (*p* < 0.01) of time × group was observed for %FM, VFL, and appendicular skeletal muscle mass. Exercise duration had an effect (*p* = 0.043) on participants’ personal and affective identification with nature, and the time × group × medication interaction significantly influenced sleep efficiency (*p* = 0.034). The exercise program proved effective in reducing total and central adiposity levels; however, it did not lead to improvements in sleep duration, sleep quality, or connection with nature.

## 1. Introduction

Regular physical activity (PA) during postmenopause plays a vital role in decreasing the intensity and frequency of climacteric symptoms, while also mitigating health risks associated with hypoestrogenism. It supports the regulation of body fat, enhances bone and muscle health, and lowers the risk of chronic conditions such as cancer, type 2 diabetes, dyslipidemia, and cardiovascular diseases [1,2]. Additionally, PA has been shown to alleviate sleep disturbances [1,2], mood instability [3,4,5], depression, and anxiety in postmenopausal women [6,7,8]. When practiced in natural environments, PA amplifies its therapeutic benefits by fostering a harmonious interplay between physical well-being and environmental engagement [9,10]. This integration promotes behaviors conducive to maintaining a healthy body weight [11], supports adherence to public health recommendations for PA (at least 150 min of moderate-to-vigorous activity per week), and enhances social connectedness [12].

Exercise programs designed for women in the climacteric phase offer not only significant physical and psychological benefits but also foster environments that encourage social interaction, the development of support networks, and sustained engagement in regular physical activity [13]. The multicomponent exercise program plays a key role in reversing frailty among older women, contributing to the reduction in fall prevalence and the decline in functional capacity [14,15,16,17]. Its benefits are well documented and include improvements in muscle strength [16,17], flexibility [16], aerobic fitness [16], balance and mobility [16,18], gait speed [16], bone mineral density [17], blood pressure [18], climacteric symptoms [19], quality of life [17,19], and arterial stiffness [20]. Training durations exceeding 12 weeks have been shown to be more effective than shorter interventions, highlighting the need for consistent practice to sustain the benefits achieved through the exercise program [16].

Despite growing interest in this area, there remains a notable gap in the literature, with few and methodologically inconsistent studies simultaneously addressing outdoor PA, individuals’ subjective connection to nature, and associated psychophysical health outcomes—particularly when employing validated instruments such as the Nature Relatedness Scale (NR-Scale) [21]. Many studies employing the NR-Scale predominantly focus on the general population [22,23,24], often neglecting specific subgroups such as postmenopausal women. This subgroup may exhibit distinct physiological and psychological responses to both physical activity and environmental exposure due to the hormonal and neuroendocrine changes associated with menopause. Addressing this gap is crucial, as nature connectedness—defined as the emotional, cognitive, and perceptual bond individuals develop with the natural environment—may significantly influence how individuals derive benefits from outdoor PA [25].

Evidence indicates that a stronger connection to nature correlates with enhanced physical and psychological well-being, which may be particularly pertinent for postmenopausal women [9]. Evaluating nature connectedness in this population may offer insight into how outdoor physical activity benefits health, guiding personalized, nature-based interventions to enhance resilience and quality of life after menopause. Therefore, this study investigated the effects of a sixteen-week green exercise program on body composition, sleep duration and quality, and nature connectedness in postmenopausal women.

## 2. Materials and Methods

### 2.1. Study Design

This quasi-experimental study is part of the community project Meno(s)Pausa+Movimento, promoted by the Municipality of Penafiel (Portugal) and scientifically and pedagogically coordinated by the University of Trás-os-Montes and Alto Douro (UTAD). Data were collected between January and June 2023. Participants were recruited through a comprehensive strategy that included personal invitations, social media campaigns, dissemination of information at healthcare facilities, and announcements in local newspapers. Prior to study enrollment, all participants underwent an initial medical evaluation to confirm eligibility criteria and collect pertinent clinical data, ensuring safe and individualized supervision throughout the exercise sessions. Six exercise groups were subsequently formed, with enrollment based on the order of registration and conditional upon medical clearance. Once these groups reached full capacity, the remaining eligible participants were allocated to the control group. Due to logistical constraints and the limited number of available spots, randomization was not feasible.

On the scheduled assessment day, participants underwent body composition analysis and completed the nature connectedness questionnaire. Subsequently, accelerometers were distributed to monitor sleep duration and quality parameters, with comprehensive instructions provided to ensure proper device usage. The preparatory guidelines for the body composition measurement were communicated to all participants in advance, and all of them had access to the results of the assessments conducted in the study.

The study strictly adhered to the ethical standards established by the Declaration of Helsinki. All procedures were thoroughly explained to potential participants, who provided written informed consent prior to enrollment (Appendix A). Ethical approval was granted by the Ethics Committee of the University of Trás-os-Montes and Alto Douro (Reference: Doc108-CE-UTAD-2022).

### 2.2. Sample

The study included 75 postmenopausal women, who were divided into two groups: experimental group (EG, n = 55, age, 57.79 ± 7.04 years; body mass, 69.07 ± 11.10 kg) and control group (CG, n = 20, age, 57.38 ± 6.47 years; body mass, 71.10 ± 11.60 kg). The recruitment period lasted from October to December 2021, during which 181 women expressed interest in participating (Figure 1). The following inclusion criteria were considered: (a) women registered in the Tâmega e Sousa Local Health Unit and residing in the municipality of Penafiel; (b) age of 42 years or older; (c) absence of premature ovarian insufficiency; (d) no hepatic, metabolic, and/or respiratory diseases that would prevent engagement in moderate-to-vigorous physical activity; (e) no symptoms of angina pectoris or myocardial infarction within the past 3 months; (f) not taking beta-blockers or antiarrhythmic medications; (g) absence of uncontrolled hypertension (systolic blood pressure ≥ 200 mmHg and/or diastolic blood pressure ≥ 105 mmHg); and (h) musculoskeletal or joint limitations that could hinder or be exacerbated by physical exercise. The exclusion criteria were: (i) non-compliance with the assessments stipulated in the program and (ii) failure to attain a minimum attendance rate of 75%.

Participants in the CG were instructed to maintain their usual activities throughout the study. During the intervention, twenty-eight participants in the EG did not meet the required attendance rate, and nine withdrew from the program for personal reasons. In the CG, loss to follow-up was observed in twenty-two women, mostly due to voluntary withdrawal (Figure 1). The final analysis included a sample of 75 women, with 55 allocated to the EG. The EG were subjected to a 16-week multicomponent exercise intervention that combines at least three types of exercise: cardiorespiratory, strength, balance, coordination, and flexibility, considered in the literature as a key strategy to maintain and improve well-being in healthy adults [16].

The study included women with natural menopause—defined as 12 consecutive months of amenorrhea not attributable to any identifiable pathological or physiological cause—as well as women with induced menopause, resulting from medical depletion of the ovarian follicular reserve. Participants encompassed both early postmenopausal women (up to six years after the final menstrual period) and late postmenopausal women (beyond this threshold), in accordance with the STRAW+10 criteria [26]. Hormone therapy use was also taken into consideration.

### 2.3. Anthropometry/Body Composition

Height was measured using an SECA 220 stadiometer (Seca Corporation, Hamburg, Germany), following the procedures described by Heyward & Wagner [27] and considering a tolerance limit of 2 mm [28]. The octopolar bioelectrical impedance device InBody 120 (Biospace, Seoul, Republic of Korea) [29] was used to assess body composition, with the following variables evaluated: body mass (BM, kg), fat mass (FM, kg and %), visceral fat level (VFL, points), trunk skeletal muscle mass (TSMM, kg), appendicular skeletal muscle mass (ASMM, kg, muscle mass of the upper and lower limbs), and appendicular skeletal muscle mass index (ASMMI = ASMM/BM^2^, kg/m^2)^. This technology employs eight contact electrodes: two positioned on the palm and thumb of each hand, and two on the anterior and posterior aspects of each foot. The device performs ten bioelectrical impedance measurements at two frequencies (20 kHz and 100 kHz) across five body segments. According to the criteria specified in the equipment manual, participants were instructed (1) not to eat food for at least 4 h; (2) not to engage in moderate to vigorous physical activity 12 h before the evaluation; (3) to use the bathroom 30 min before the test (to reduce the volume of urine and feces); (4) not to consume alcoholic beverages for at least 48 h; (5) not to wear metal jewelry [27]. Data were imported electronically into spreadsheets using Lookin’Body 120 software (Biospace, Seoul, Republic of Korea). The cut-off points for elevated visceral fat level (VFL), obesity, and low muscle condition were, respectively: VFL > 9 points [29], FM ≥ 35% [30], and ASMMI < 5.5 kg/m^2^ [31].

### 2.4. Sleep

The triaxial accelerometer ActiGraph GT3X (ActiGraph LLC, Pensacola, FL, USA) was employed to assess the following objective sleep measures: total sleep time (TST, hours/night), sleep efficiency (SE, %, relationship between the total sleep time and the total time each woman spent in bed attempting to sleep); nocturnal awakenings (NA, number; number of sleep interruptions that occur during the night when the woman briefly awakens after having fallen asleep), time of awakenings (MA, minutes), and sleep fragmentation index (SFI, number of events, total number of awakenings divided by total sleep time in hours; it assesses the frequency and duration of awakenings or movements that fragment sleep, reflecting the extent of sleep disruption).

All participants received standardized oral and written instructions on how to use the device. The accelerometer was worn on the non-dominant wrist for four nights (two weekdays and two weekend days). Sleep patterns were analyzed using a previously validated software algorithm based on the Cole–Kripke scoring method [32]. The TST was categorized as sufficient (≥7 h per night) or insufficient (<7 h per night), following the guidelines established by Cole et al. [32]. Sleep efficiency was classified as low (<85%) or high (≥85%) based on the thresholds defined by Lacks and Morin [33].

Furthermore, the recommended cutoff for the SFI is fewer than 5 events per hour [3]. The data were processed and analyzed using ActiLife software (v6.13.4, Pensacola, FL, USA). The use of pharmacological agents with potential effects on sleep duration and quality was assessed, including benzodiazepines, non-benzodiazepine hypnotics, sedative antidepressants, sedative antipsychotics, melatonin and melatonin receptor agonists, activating antidepressants, corticosteroids, beta-blockers, decongestants, and bronchodilators. Information regarding medication use was obtained during a medical evaluation conducted prior to participants’ enrollment in the study.

### 2.5. Connection with Nature

The participants’ relationship with the natural environment was assessed using the Nature Relatedness Scale (NR-Scale), developed by Nisbet et al. [1] and validated for the Portuguese population by Conceição [34]. This scale comprises 21 items rated on a 5-point Likert scale (1 = very low, 5 = very high), enabling the evaluation of three dimensions: NR-Self, which measures identification with nature (average score of items 5, 7, 8, 12, 14, 16, 17, and 21); NR-Perspective, which assesses concern about the impact of human actions on the environment (average score of items 2, 3, 11, 15, 18, 19, and 20); and NR-Experience, which reflects an individual’s comfort in nature and desire to engage with it (average score of items 1, 4, 6, 9, 10, and 13). The total NR score was also calculated. The questionnaire was administered by the same researcher, who addressed any participant questions without influencing their responses.

### 2.6. The Program Meno(s)Pausa+Movimento

The Meno(s)Pausa+Movimento is a multicomponent exercise program that prioritizes physical activity in contact with the natural environment. The training program incorporated cardiorespiratory, resistance, neuromotor (or functional), and flexibility exercises (Figure 2) and was delivered through three 60 min sessions per week on non-consecutive days [35]. This training is designed to engage multiple muscle groups in a coordinated and efficient manner, delivering diverse physiological stimuli that promote broad functional adaptations. Each session began with a warm-up of approximately 8 min, including both general and specific movements, as well as static stretching.

Cardiorespiratory work was incorporated in every session, lasting about 30 min, and was carried out in a public garden in the municipality of Penafiel (Portugal). This garden features a significant diversity of shrub and tree species, whose interaction with the romantic-style historical heritage, along with the presence of several channels and lakes, creates a notable aesthetic impact (Figure 3). Various groundcover plants, such as grasses and bulbous species, together with diverse flowerbeds, contribute substantially to the beauty of the space, providing a natural and tranquil environment that promotes the physical and psychological well-being [36]. Cardiorespiratory training was incorporated into all sessions, comprising walking and running activities, with a progressive increase in both duration (from 20 to 30 min) and exercise intensity (from 30–40% to 60–75% of heart rate reserve) (Appendix A).

Resistance training was conducted twice a week, focusing on large muscle groups and body regions most susceptible to sarcopenia and osteoporosis [35]. To optimize functional capacity and reduce the risk of falls, strength training (ranging from 40–50% to 80% of 1RM (repetition maximum); 2 sets of 8–12 repetitions) was combined with power training, the latter adjusted with different intensity levels for upper and lower limbs (Appendix A). Strength training was specifically aimed at promoting muscle hypertrophy, while power training focused on developing the ability to generate force rapidly, a crucial factor for fall prevention, gait improvement, reduction in reaction time, and optimization of performance in functional activities [37].

Throughout the sessions, different myofascial meridians were explored to promote functional and global movement integration. The program included one weekly neuromotor exercise session lasting approximately 10 min, performed at moderate to vigorous intensity, 40–75% of heart rate reserve (HRR), focusing on motor coordination, balance, agility, and proprioception (Figure 2). The flexibility component was also conducted once a week, emphasizing static stretching and prioritizing the main muscle and tendon groups [35]. Each session concluded with a cooldown phase lasting 8 to 10 min, consisting of static stretching exercises focused primarily on the muscle groups most engaged during the main part of the session, accompanied by a gradual reduction in exercise intensity. During the exercise sessions, intensity was monitored using heart rate, measured with an H10 heart rate monitor (Polar Electro, Kempele, Finland), or the Borg Rating of Perceived Exertion (RPE) scale (6 to 20). All exercise sessions were supervised by qualified professionals, with a minimum attendance rate of 75% required.

### 2.7. Statistical Analysis

All analyses were performed using IBM SPSS Statistics, version 27.0 (IBM Corp., Armonk, NY, USA). A *p*-value ≤ 0.05 was considered statistically significant. Continuous variables were expressed as mean ± standard deviation, while categorical variables were presented as absolute frequencies and percentages. Variable change rates were calculated for both groups. To compare baseline values and sample characteristics between the EG and CG, independent samples *t*-tests were used for normally distributed data, and the Mann–Whitney U test was applied for non-normally distributed data. A two-way mixed-design ANOVA was conducted to examine the main and interaction effects of exercise duration and group on changes in body composition and nature connectedness. For the sleep-related variables, a three-way ANOVA was performed, incorporating medication as a factor due to its potential influence on these outcomes. Effect sizes for the independent variables on the dependent variables were assessed using partial η^2^, and observed power was calculated to evaluate the reliability of the results. The minimum required sample size was estimated using G*Power 3.1.9.7, based on an independent samples *t*-test. A large effect size (d = 0.80), a significance level of 5% (α = 0.05), and a statistical power of 80% (1 − β = 0.80) were specified. The calculation indicated a minimum of 14 participants in one group and 38 participants in the other, reflecting the ratio adopted for the study design.

## 3. Results

The characteristics of the participants are summarized in Table 1 and Table 2. Most participants had experienced natural menopause (89.1% in the EG and 100% in the CG), had not used hormone therapy (80% and 85%, respectively), and had been postmenopausal for more than six years (50.9% and 55.0%). The difference at baseline between the two groups was not identified. When the modification rates of the variables related to connection with nature were compared, no statistically significant differences were observed between the groups. The EG showed improvements in the total score of the Nature Relatedness Scale, as well as across its different dimensions, with notable increases in NR-Self (+4.20%) and NR-Experience (+3.03%). In the CG, no improvements were observed in the dimension related to sustainability values and the perception of the impact of human actions on the environment, with a slight decrease (−1.32%).

Both groups showed a reduction in TST (−7.23% in the EG and −11.86% in the CG) and SE (−3.56% and −2.59%, respectively), as well as an increase in the SFI (+25.45% and +7.39%, respectively). However, when comparing the rates of change in these variables between the two groups, statistically significant differences (*p* ≤ 0.05) were observed only for the SFI (Table 1).

After 16 weeks of intervention, 20% of the participants in the EG and 10% in the CG showed signs suggestive of possible sleep disorders (Table 2).

The EG showed a reduction in BM (−2.53%), supported by a decrease in absolute fat mass (−9.26%) and, more notably, a reduction in VFL (−13.46%). In the CG, a decrease in adiposity levels was also observed, although to a lesser extent (−1.21% in %FM and −3.80% in VFL). Statistically significant differences (*p* ≤ 0.01) were identified between the two groups (Table 1). After 16 weeks of intervention, the percentage of women with total and central obesity in the experimental group (EG) decreased from 76.4% to 65.5% and from 80% to 69.1%, respectively (Table 2). At the start of the study, four participants in the experimental group had low muscle mass. However, there were no significant differences in the changes in muscle mass of the trunk and limbs between the experimental and control groups.

In Table 3, a two-way mixed ANOVA with the variables of connection with nature and body composition can be observed. A significant interaction effect between time and group (time × group; *p* = 0.001) was identified for %FM, VFL, and ASMM, with 13.5% to 13.8% of the variance in the response outcome explained by the interaction, and an observed power greater than 0.90. Additionally, the time had a significant influence on ASMMI (*p* < 0.001), with a partial η^2^ of 0.325. The duration of the intervention had a statistically significant effect on the NR-Self dimension (*p* = 0.043), with a moderate effect size (partial η^2^ = 0.055) and an observed power of 0.529, indicating a moderate probability of detecting a true effect, given the sample size and data variability.

Regarding sleep efficiency, the combined effect of intervention time (baseline vs. 16 weeks) and group (experimental vs. control) was a significant effect by the medication used by the participants (*p* = 0.034), suggesting that the impact of the intervention varied according to the use of drugs with potential effects on sleep (partial η^2^ = 0.062; observed power 0.571).

## 4. Discussion

This study examined the effects of a 16-week multimodal green exercise program on body composition, sleep, and nature connectedness in postmenopausal women. The exercise program proved effective in reducing total adiposity levels, particularly central adiposity, showing a significantly different progression between groups over time. The intensity achieved during cardiorespiratory exercise may have been a determining factor in the results observed. Evidence suggests that aerobic exercise performed at moderate to vigorous intensity promotes more efficient mobilization of lipid reserves, particularly visceral fat [38]. This adipose region is metabolically active and highly responsive to physical exercise stimuli due to specific cellular characteristics, such as smaller adipocyte volume, greater vascularization and innervation, reduced sensitivity to anti-lipolytic receptors (such as α2-adrenergic receptors), and increased sensitivity to β-adrenergic receptors, thereby enhancing lipolysis [39]. Additionally, performing cardiorespiratory exercise in natural environments may have further amplified the health benefits observed. Physical activity conducted in green settings has been associated with increased enjoyment, reduced perceived exertion, and greater adherence to exercise programs [12]. Growing evidence highlights access to green spaces as a significant environmental determinant of health. Specifically, environments characterized by high plant diversity [40,41], substantial tree canopy cover [41], and infrastructure that facilitates walking and running [42] play a crucial role in promoting physical activity and supporting the maintenance of a healthy body weight. These natural features not only enhance aesthetic and sensory experiences but also contribute to more favorable microclimatic conditions and promote consistent engagement with outdoor environments, thereby improving metabolic health outcomes.

Muscle mass loss accelerates after the age of 60, particularly affecting the lower limbs [43,44]. The decline in muscle strength is two to five times greater than the reduction observed in muscle size [45], and the decrease in muscle power appears to have an even more significant impact on the performance of activities of daily living compared to muscle strength alone [46]. The decline in estradiol levels associated with menopause impairs the activation and proliferation of satellite cells (muscle stem cells), mediated through specific estrogen receptors, thereby compromising the preservation and regeneration of muscle tissue [47,48]. This hormonal reduction contributes to an estimated annual loss of approximately 0.6% of muscle mass in women [49], with a more pronounced decline typically observed between the ages of 60 and 69, and a marked acceleration from the age of 70 onwards [31]. In the present study, sarcopenia was identified in only four participants at baseline, all of whom belonged to the experimental group (EG). Following the 16-week intervention, this number decreased. The low prevalence of sarcopenia observed in this sample may be partly explained by its age distribution (32% of participants were between 60 and 69 years old, and only 4% were aged 70 or above) and partly by methodological limitations of the muscle mass assessment tool employed. Although an eight-electrode bioimpedance device was used, its measurement capacity was restricted to a frequency range up to 100 kHz. In contrast, more advanced bioimpedance devices operating at multiple frequencies between 1000 and 3000 kHz offer greater precision in distinguishing body compartments and in the estimation of skeletal muscle mass. Although the EG and CG did not show statistically significant differences in the rates of change in ASMM, the significant time × group interaction indicates that the response to the intervention varied according to group, despite similar final outcomes.

The results indicated that no statistically significant differences were found in the rates of change in ASMM between groups over the 16-week intervention, the progression of this body composition variable differed, with the control group (CG) showing more positive improvements, contrary to initial expectations. This observation may be attributed to several factors. In the first place, despite the progression in intensity, the intervention protocol consisted of only two short sessions per week (20 min each), which may have been insufficient to induce significant adaptations in this population. The literature suggests that, especially in postmenopausal women, higher training volumes and increased weekly frequency are generally more effective in promoting gains in muscle mass and strength [50]. On the other hand, the possibility cannot be ruled out that spontaneous lifestyle changes in the control group, such as increased non-structured physical activity, dietary modifications, or reduced sedentary behavior, contributed to the observed results.

Data analysis also revealed a significant main effect of time on the variation in ASMMI, with improvements observed over the 16-week period in both groups. Although the time × group interaction did not reach statistical significance, a trend toward a greater increase was observed in the experimental group, suggesting a possible additional benefit associated with the implemented intervention.

Sleep disturbances affect more than 50% of postmenopausal women and are generally attributed to multifactorial causes [12]. Insomnia, with or without comorbidities such as anxiety or mild depression, is the most prevalent manifestation in this group. Obesity [51] and the redistribution of body fat to central regions, including the neck, tend to exacerbate pharyngeal collapse during sleep, compromising nocturnal oxygenation and leading to greater sleep fragmentation [52]. Although the majority of participants in our study presented clinically significant excess weight, the indicators related to sleep duration and quality were generally favorable at baseline. This finding may be associated with the fact that 92% of the sample had undergone natural menopause, a condition often associated with a lower incidence of sleep disorders compared to induced menopause [53]. However, the duration of the intervention was associated with a decline in sleep quality, as evidenced by significant effects on the SFI and a reduction in total sleep time. These results may reflect a complex interaction between physiological and behavioral factors. The reduction in estrogen levels impairs serotonin synthesis, contributing to mood alterations and decreasing melatonin production by the pineal gland. Simultaneously, estrogen deficiency tends to affect thermoregulation, causing symptoms such as hot flashes and night sweats, which together reduce REM sleep, decrease total sleep duration, and increase sleep fragmentation [54]. The decrease in progesterone levels reduces the activation of benzodiazepine receptors and gamma-aminobutyric acid (GABA) receptors, limiting their anxiolytic effects and promoting states of neural hyperexcitation that contribute to sleep disturbances. Moreover, lifestyle changes—even when beneficial in the medium and long term—can initially disrupt circadian rhythms, especially in postmenopausal women who are more vulnerable to sleep disturbances. However, these disruptions tend to be transient and are often followed by adaptation processes and gradual improvements in sleep quality [55]. During this transitional phase, symptoms such as hot flashes, musculoskeletal pain, irritability, and anxiety may intensify, exacerbating sleep fragmentation and impairing sleep depth [56]. It is also important to highlight that physical activity interventions, despite their well-established benefits on sleep, can initially worsen sleep fragmentation—particularly when there are significant changes in intensity, frequency, or timing of exercise [57,58]. Although greater exposure to green spaces is associated with improvements in sleep quality and duration [59], the duration of the intervention conducted in an urban green space may have been insufficient to induce significant improvements in sleep parameters. Thus, the initial deterioration in sleep quality may reflect a transient process of physiological and behavioral adaptation to the newly introduced routine. Therefore, prolonged monitoring becomes essential to assess whether this trend reverses or stabilizes over time.

Regarding sleep efficiency, the results suggest that, over the course of the intervention, this parameter evolved differently between the two groups, with a trend toward a more pronounced reduction in the experimental group compared to the control group, although this difference did not reach statistical significance. This variation appears to depend on the use of medications with an impact on sleep. It is expected that participants using sedative or sleep-regulating medication exhibited different SE levels compared to those who did not, reflecting both the intended therapeutic effects of these drugs and the presence of more severe sleep disturbances that may have prompted their prescription [60,61]. Medication can contribute to greater ease in initiating sleep and to reducing nocturnal awakenings, which tends to improve sleep efficiency [62]. However, in some cases, the prolonged or inappropriate use of hypnotics may lead to superficial or fragmented sleep, ultimately compromising sleep quality even when efficiency appears preserved [63]. Environments containing natural elements provide favorable contexts for the promotion of health and well-being by encouraging regular physical activity, facilitating psychological restoration, reducing stress levels, and promoting social connection and a sense of belonging [64,65]. In postmenopausal women, connection with nature appears to be particularly beneficial in managing climacteric symptoms, while also contributing to the strengthening of identity and life purpose. This reinforcement is especially relevant, given that the menopausal experience involves not only physiological changes but also a reassessment of the traditional social and familial roles attributed to women, which can negatively affect self-image and self-esteem [66].

In the present study, no statistically significant differences were observed in the variation in the Nature Relatedness Scale and its dimensions between the groups analyzed. Nonetheless, the EG showed a consistent trend toward greater improvements compared to the CG, particularly in the NR-Self dimension. According to McNeil et al. [67], engaging in physical activity in natural environments fosters the development of a stronger connection with the natural world, which, in turn, is associated with higher levels of social connection. The study by Sheffield et al. [68] suggests that performing cardiovascular exercise in natural settings three times a week may have contributed to the outcomes observed in the experimental group. However, these effects might have been more pronounced if the duration of nature exposure had been longer (more than 20–30 min per session), allowing for deeper observation and repeated engagement with the natural environment. Additionally, the presence of greater biodiversity in the outdoor spaces could also have intensified the participants’ sense of closeness and connection to nature.

Data analysis further revealed that, regardless of group assignment, the intervention period had a significant impact on the NR-Self dimension. This suggests that participants began to more deeply incorporate nature into their self-concept, perceiving it as a central component of their personal identity [1].

This study presents some methodological limitations that should be considered when interpreting the results. First, the small sample size limits the statistical power of the analysis and restricts the generalizability of the findings to other populations of postmenopausal women. In addition, participants were not randomly selected, which may compromise the representativeness of the sample in relation to the general population.

Finally, the use of bioelectrical impedance analysis to assess body composition represents an additional limitation, as this method is not considered the gold standard for such measurements. The use of more accurate techniques, such as dual-energy X-ray absorptiometry (DXA), could have provided a more precise assessment.

## 5. Conclusions

The exercise program demonstrated effectiveness in reducing both total and central adiposity among participants. However, the 16-week intervention did not result in significant changes in participants’ connection with nature, nor in sleep duration or efficiency. Nonetheless, an increase in the number of awakenings and transitions between sleep stages was observed in the EG compared to the CG, suggesting more fragmented and potentially less restorative sleep. These findings suggest that additional studies may be needed to determine whether other specific features of the natural environment may be more likely to improve sleep variables and connection with nature. This study is a pioneer in the implementation of a multicomponent green exercise program, assessing body composition, sleep parameters, and connection with nature, specifically in postmenopausal women. Given the unique physiological, metabolic, and psychosocial changes that occur during this phase of life, the evaluation of these results in this population is of critical importance for the development of effective and adapted health strategies.

## Figures and Tables

**Figure 1 ijerph-22-01216-f001:**
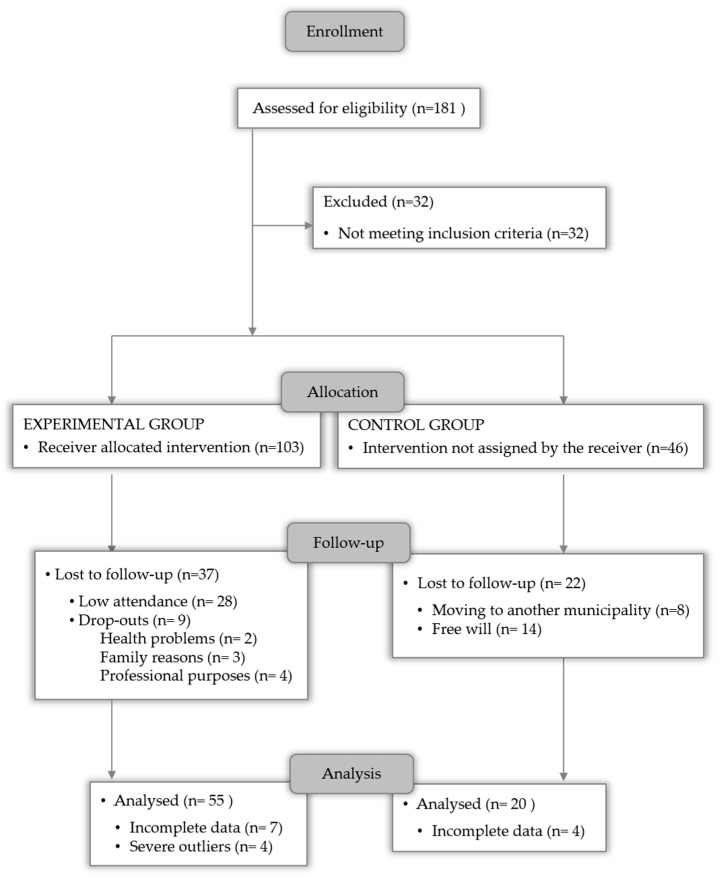
Meno(s)Pausa+Movimento flowchart.

**Figure 2 ijerph-22-01216-f002:**
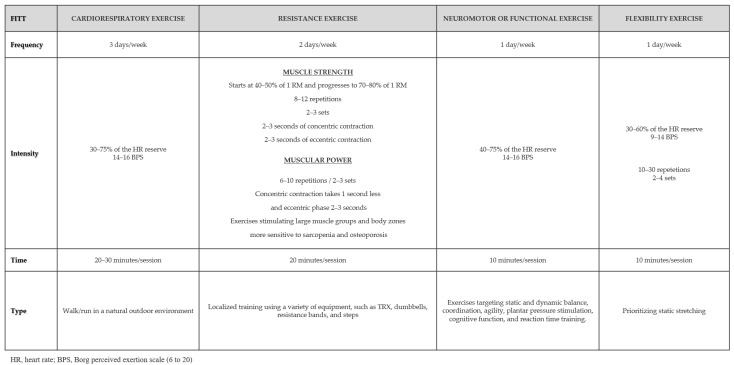
Structure of the multicomponent training program implemented in Meno(s)Pausa+Movimento [35].

**Figure 3 ijerph-22-01216-f003:**
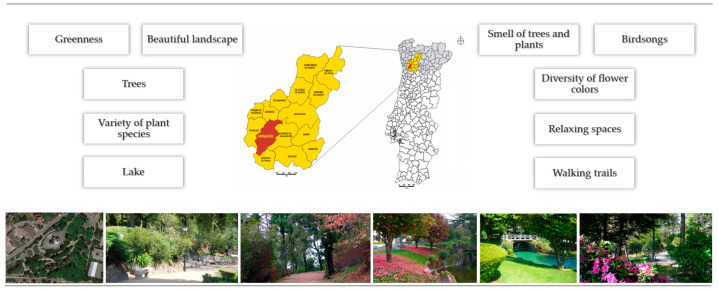
Sameiro Public Garden, located in the municipality of Penafiel, where the cardiorespiratory exercise of the Meno(s)Pausa+Movimento program was carried out.

**Table 1 ijerph-22-01216-t001:** Characterization of the two groups under analysis and differences between them.

Variables	Experimental Group (n = 55)			Control Group (n = 20)		
	Baseline Mean ± SD	16 Weeks Mean ± SD	Δ %	Baseline Mean ± SD	16 Weeks Mean ± SD	Δ %
Body Composition						
Body mass (BM, kg)	69.07 ± 11.10	67.14 ± 10.03	−2.53 ± 4.52	71.10 ± 11.60 ^(a)^	71.67 ± 12.70	0.62 ± 2.98 ** ^(b)^
Height (H, m)	1.57 ± 0.05	1.57 ± 0.05	−0.11 ± 0.97	1.58 ± 0.05 ^(a)^	1.58 ± 0.05	0.24 ± 0.58 ^(b)^
Fat mass (FM, kg)	28.18 ± 8.84	25.48 ± 8.20	−9.26 ± 9.89	29.44 ± 8.79 ^(b)^	29.23 ± 9.39	−1.21 ± 6.21 ** ^(b)^
Fat mass (FM, %)	39.97 ± 6.81	37.15 ± 7.06	−7.16 ± 7.00	40.69 ± 6.38 ^(a)^	39.98 ± 6.74	−1.87 ± 3.80 ** ^(b)^
Visceral fat level (VFL, points)	13.60 ± 4.42	11.80 ± 4.37	−13.46 ± 11.62	14.20 ± 4.64 ^(b)^	13.75 ± 4.82	−3.80 ± 6.91 ** ^(b)^
Trunk skeletal muscle mass (TSMM, kg)	18.85 ± 2.11	18.76 ± 1.96	−0.36 ± 3.07	19.26 ± 2.17 ^(b)^	19.37 ± 2.37	0.52 ± 2.83 ^(a)^
Appendicular skeletal muscle mass (ASMM, kg)	16.14 ± 2.01	16.55 ± 1.98	2.64 ± 3.71	16.51 ± 2.37 ^(a)^	17.0 ± 2.62	3.22 ± 3.87 ^(a)^
Appendicular skeletal muscle mass index (ASMMI kg/m^2^)	6.49 ± 0.64	6.68 ± 0.61	3.01 ± 4.17	6.61 ± 0.68 ^(a)^	6.80 ± 0.81	2.73 ± 3.67 ^(b)^
Sleep						
Total sleep time (TST, Hrs/night)	7.46 ± 1.01	6.81 ± 0.82	−7.23 ± 16.17	7.57 ± 1.05 ^(a)^	6.60 ± 1.02	−11.86 ± 14.63 ^(b)^
Sleep efficiency (SE, %)	90.84 ± 4.14	87.47 ± 5.48	−3.56 ± 6.81	89.48 ± 2.94 ^(a)^	87.10 ± 6.66	−2.59 ± 7.61 ^(b)^
Nocturnal awakenings (NA, No.)	14.48 ± 5.88	15.56 ± 5.08	25.10 ± 86.87	16.70 ± 4.44 ^(b)^	17.78 ± 9.50	7.54 ± 45.35 ^(b)^
Minutes of awakenings (MA, Min)	2.89 ± 0.78	3.51 ± 1.21	28.62 ± 53.28	3.13 ± 0.78 ^(a)^	3.91 ± 2.66	28.24 ± 86.20 ^(b)^
Sleep fragmentation index (SFI, No.)	3.10 ± 1.01	3.66 ± 1.20	25.45 ± 44.73	3.56 ± 0.69 ^(a)^	3.64 ± 1.48	7.39 ± 55.35 * ^(b)^
Nature Relatedness						
NR-Self (points)	4.35 ± 0.37	4.51 ± 0.26	4.20 ± 9.27	4.34 ± 0.32 ^(a)^	4.41 ± 0.32	1.93 ± 9.90 ^(b)^
NR-Perspective (points)	2.73 ± 0.39	2.75 ± 0.38	2.33 ± 18.43	2.91 ± 0.44 ^(a)^	2.88 ± 0.54	−1.32 ± 10.62 ^(b)^
NR-Experience (points)	3.28 ± 0.41	3.35 ± 0.43	3.03 ± 15.23	3.37 ± 0.47 ^(b)^	3.36 ± 0.39	1.59 ± 17.94 ^(a)^
NR-Scale (points)	3.50 ± 0.23	3.59 ± 0.24	2.61 ± 7.11	3.59 ± 0.26 ^(a)^	3.60 ± 0.27	0.54 ± 7.55 ^(a)^

^(a)^ *t*-test independent samples; ^(b)^ Mann–Whitney test; Absence of differences between the baseline values of the two groups; * *p* ≤ 0.05; ** *p* ≤ 0.01.

**Table 2 ijerph-22-01216-t002:** Frequency of variables in the experimental group and control group.

**Variables**	**Experimental Group (n = 55)**		**Control Group (n = 20)**	
	**n (%)**		**n (%)**	
Nature of menopause				
Natural	49 (89.1)		20 (100.0)	
Induced	6 (10.9)		0 (0.0)	
Hormone therapy				
No	44 (80.0)		17 (85.0)	
Yes	11 (20.0)		3 (15.0)	
Stage of menopause				
Early postmenopause	27 (49.1)		9 (45.0)	
Late postmenopause	28 (50.9)		11 (55.0)	
**Variables**	**Experimental Group (n = 55)**		**Control Group (n = 20)**	
	**Baseline** **n (%)**	**16 weeks** **n (%)**	**Baseline** **n (%)**	**16 weeks** **n (%)**
Obesity				
No	13 (23.6)	19 (34.5)	5 (25.0)	5 (25.0)
Yes	42 (76.4)	35 (65.5)	15 (75.0)	15 (75.0)
Central obesity				
No	11 (20.0)	17 (30.9)	2 (10.0)	6 (30.0)
Yes	41 (80.0)	38 (69.1)	18 (90.0)	14 (70.0)
Muscle condition				
Normal	51 (92.7)	53 (96.4)	20 (100.0)	19 (95.0)
Low	4 (7.3)	2 (3.6)	0 (0.0)	1 (5.0)
Sleep duration				
Sufficient sleep	42 (76.4)	27 (49.1)	15 (75.0)	7 (35.0)
Insufficient sleep	13 (23.6)	28 (50.9)	5 (25.0)	13 (65.0)
Sleep efficiency				
Recommended	50 (90.9)	41 (74.5)	20 (100.0)	15 (75.0)
Not recommended	5 (9.1)	14 (25.5)	0 (0.0)	5 (25.0)
Sleep fragmentation index				
Normal	53 (96.4)	44 (80)	20 (100)	18 (9)
Possible sleep disturbance	2 (3.6)	11 (20)	0 (0)	2 (10)

**Table 3 ijerph-22-01216-t003:** Two-way mixed ANOVA for body composition and connection with nature and three-way mixed ANOVA for sleep variables.

Dependent Variables	Two-Way Mixed ANOVA											
	Time				Time × Group				Time × Group × Medication			
	F	*p*	Partial η^2^	Observed Power	F	*p*	Partial η^2^	Observed Power	F	*p*	Partial η^2^	Observed Power
Body Composition												
Fat mass (FM, %)	31.672	<0.001	0.303	1.000	11.410	0.001	0.135	0.915				
Visceral fat level (VFL, points) ^µ^	31.931	<0.001	0.304	1.000	11.495	0.001	0.136	0.917				
Trunk skeletal muscle mass (TSMM, kg)	0.018	0.893	<0.001	0.052	1.648	0.203	0.022	0.245				
Appendicular skeletal muscle mass (ASMM, kg) ^β^	33.575	<0.001	0.315	1.000	11.641	0.001	0.138	0.920				
Appendicular skeletal muscle mass index (ASMMI kg/m^2^)	35.185	<0.001	0.325	1.000	0.725	0.397	0.010	0.134				
Sleep												
Total sleep time (TST, Hrs/night)	25.542	<0.001	0.265	0.999	1.939	0.169	0.026	0.278	1.048	0.310	0.015	0.172
Sleep efficiency (SE, %) ^α^	17.976	<0.001	0.202	0.987	0.186	0.668	0.003	0.071	4.697	0.034	0.062	0.571
Nocturnal awakenings (NA, No) ^γ^	1.599	0.210	0.022	0.239	0.871	0.354	0.012	0.151	0.046	0.830	0.001	0.055
Minutes of awakenings (MA, Min) ^χ^	7.004	0.010	0.090	0.742	0.300	0.586	0.004	0.084	0.102	0.750	0.001	0.062
Sleep fragmentation index (SFI, No)	9.010	0.004	0.113	0.842	0.063	0.802	0.001	0.057	3.189	0.078	0.043	0.422
Nature Relatedness												
NR-Self (points) ^α^	4.246	0.043	0.055	0.529	0.744	0.391	0.010	0.136				
NR-Perspective (points) ^µ^	0.025	0.874	<0.001	0.053	0.240	0.625	0.003	0.077				
NR-Experience (points) ^ε^	0.119	0.731	0.002	0.063	0.371	0.544	0.005	0.092				
NR-Scale (points)	2.103	0.151	0.028	0.299	1.265	0.264	0.017	0.199				

η^2^ partial-Partial Eta Squared; ^α^, quadratic transformation to improve distribution asymmetry; ^ε^, quadratic transformation to normalize distribution; ^χ^, reciprocal square root transformation to improve distribution asymmetry; ^γ^, logarithmic transformation to normalize distribution; ^β^, square root transformation to improve distribution asymmetry; ^µ^, several transformations of the variable were tested, but they did not improve the asymmetry of the distribution when compared to analyzing the same variable without transformation.

## Data Availability

The original contributions presented in this study are included in the article/Appendix A. Further inquiries can be directed to the corresponding author.

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
