# Peer review of "Effects of a 16-Week Green Exercise Program on Body Composition, Sleep, and Nature Connection in Postmenopausal Women"

_ijerph, 2025, doi:10.3390/ijerph22081216_

Round 1

Reviewer 1 Report

Comments and Suggestions for Authors

Major Comments:

  1. Could the authors clarify the rationale behind the initial allocation of participants, with n=103 in the experimental group and n=46 in the control group? What guided this uneven group size?
  2. The manuscript states that participants were non-randomly allocated. Please elaborate on the reasoning behind this decision. What constraints or considerations led to choosing a non-random design?
  3. More detail is needed regarding the design of the multicomponent training program. Why was this particular combination of activities selected? Why not focus on a single activity to more clearly assess its specific effects on health outcomes?
  4. During the 16-week intervention, what exactly did the control group do? Were they completely inactive, or did they participate in any alternative activities? Did they have any exposure to the garden environment, or were they not exposed to either the garden or physical activity?
  5. Please provide more explanation of the dependent variables used in the study. For example, what is the difference between appendicular skeletal muscle mass and appendicular skeletal muscle mass index? What health aspects do these variables reflect? Similarly, more context for the sleep-related measures would help readers who may not be experts in sports or health sciences.
  6. Much of the content in the Discussion section could be better used to frame the rationale for the study design earlier in the manuscript. Consider reorganizing this material to improve the flow and strengthen the justification for the intervention.

Detailed Comments:

  • Line 80: Should "during" be replaced with "after"?
  • Line 111: Should "failure" be changed to "success"?

Author Response

Could the authors clarify the rationale behind the initial allocation of participants, with n=103 in the experimental group and n=46 in the control group? What guided this uneven group size?

We appreciate the question raised by the reviewer.

The Meno(s)Pausa+Movimento program is supported by the Municipality of Penafiel (Portugal), which provided the necessary facilities, as well as material and technical resources for the implementation of the activities, including supervised exercise sessions and multidimensional assessments. This institutional partnership enabled the free participation of all individuals enrolled in the program.

A total of six exercise classes were formed, with enrolment based on the order of registration and conditional on medical clearance following an initial screening. Once the available exercise classes were filled, the remaining eligible participants were assigned to the control group (CG), who only took part in the assessment procedures during the intervention period. This allocation strategy, determined by logistical and contextual constraints, accounts for the larger number of participants in the experimental group (EG) compared to the CG, and reflects the non-randomized nature of the study design.

This clarification has been incorporated into the revised Methods section of the manuscript in response to the reviewer’s comment.

Please see page 3, from line 100 to line 107

The manuscript states that participants were non-randomly allocated. Please elaborate on the reasoning behind this decision. What constraints or considerations led to choosing a non-random design?

Thank you for pointing this out. The explanation is provided in the previous section.

More detail is needed regarding the design of the multicomponent training program. Why was this particular combination of activities selected? Why not focus on a single activity to more clearly assess its specific effects on health outcomes?

Multicomponent exercise programs, which integrate cardiorespiratory, resistance, flexibility, and neuromotor (or functional) training, have demonstrated significant benefits for postmenopausal women. This population often experiences a progressive decline in muscle mass, bone density, balance, and functional capacity, as well as increased risk of falls and cardiometabolic disorders due to hormonal changes associated with menopause. Evidence indicates that multicomponent interventions are effective in improving muscular strength, aerobic capacity, postural control, gait speed, and flexibility, while also contributing to the maintenance of bone health and the reduction of fall risk. Moreover, these programs have been associated with improved psychological well-being, and quality of life. We sought to clarify the importance of multicomponent exercise in this population by explicitly outlining its benefits in the Introduction section of the article. Additionally, supplementary information was included in the Methods section. 

Please see page 2, from line 63 to line 71 and Please see page 7, from line 216 to line 221

During the 16-week intervention, what exactly did the control group do? Were they completely inactive, or did they participate in any alternative activities? Did they have any exposure to the garden environment, or were they not exposed to either the garden or physical activity?

The control group did not participate in any alternative activities but, similar to the experimental group, had access to the assessment results. since the Jardim do Sameiro is a public space located in the city of Penafiel, it was not possible to guarantee that control group participants had no exposure to the garden environment.

Please provide more explanation of the dependent variables used in the study. For example, what is the difference between appendicular skeletal muscle mass and appendicular skeletal muscle mass index? What health aspects do these variables reflect? Similarly, more context for the sleep-related measures would help readers who may not be experts in sports or health sciences.

We appreciate the opportunity to explain this section of the Methods.

Appendicular skeletal muscle mass (ASMM) refers to the muscle mass of the upper and lower limbs, which are the primary regions responsible for locomotion and the performance of daily functional tasks. The appendicular skeletal muscle mass index (ASMMI) is calculated by adjusting ASMM for height squared (kg/m²), enabling standardized comparisons across individuals of varying body sizes. ASMMI is a widely accepted parameter in both clinical and research settings and is a key diagnostic criterion for sarcopenia, as recommended by the European Working Group on Sarcopenia in Older People (EWGSOP).

We sought to provide a clearer explanation of the sleep quality variables within the main body of the text.

Please see page 6, line 163

Much of the content in the Discussion section could be better used to frame the rationale for the study design earlier in the manuscript. Consider reorganizing this material to improve the flow and strengthen the justification for the intervention.

Good point. We incorporated some of the information originally presented in the Discussion section into the Introduction and Methods sections, as suggested by the reviewer, with the aim of improving the organization of the text, enhancing the flow of reading, and strengthening the study’s argumentation in a clearer and more coherent manner.

Line 80: Should "during" be replaced with "after"?

We thank the reviewer for this valuable comment. The suggested change has been incorporated into the manuscript accordingly.

Please see page 3, line 88

Line 111: Should "failure" be changed to "success"?

Thank you for this helpful comment. We have made the corresponding change in the manuscript as suggested.

Please see page 4, line 134

The introduction can be improved - We thank the reviewer for this valuable comment. In response, we have revised the introduction to better highlight the benefits of multicomponent exercise and to emphasize the importance of implementing such interventions over a period longer than 12 weeks to ensure sustained health benefits.

The study design can be improved - Thank you for this helpful comment. In the study design, we detailed the formation of the experimental and control groups, providing justification for the difference in the number of participants between them. Additionally, we described the organization of the assessments conducted throughout the study.

The Methods section can be improved - We thank the reviewer for the comment. We have made the necessary changes in the main text, as indicated in the previous points.

Reviewer 2 Report

Comments and Suggestions for Authors

I appreciate the opportunity to review this manuscript.

The paper discusses a relevant topic. This study aimed to investigate the effects of a sixteen-week green exercise program on body composition, sleep duration and quality, and nature connectedness in postmenopausal women. The paper is well written. The title clearly describes the article. The abstract reflects the content of the article. However, my concern is about the results and the discussion about sleep. This section must be improved.

Some suggestions are given to improve the manuscript.

Page 224-225: “For the sleep-related variables, a three-way ANOVA was performed, incorporating medication as a factor due to its potential influence on these outcomes.” This information appears only in statistical analysis without mentioning how these data were obtained.

Page: 274-277: “Regarding sleep efficiency, the combined effect of intervention time (baseline vs. 16 weeks) and group (experimental vs. control) was a significant effect by the medication used by the participants (p=0.034), suggesting that the impact of the intervention varied according to the use of drugs with potential effects on sleep (partial η² = 0.062; observed power 0.571).” What drugs are the authors referencing?

The results in the tables are difficult to see due to the low resolution. This must be improved.

The sleep fragmentation index (table 1) increased in GE. And this result is not described in the results section. This section must be improved.

In pages 257-259 “After 16 weeks of intervention, 20% of the participants in the EG and 10% in the CG showed signs suggestive of possible sleep disorders (Table 2).” Are these results statistically different?

The author did not answer the study objective. The authors afirmed in abstract page 43-45: “ho-ever, it did not lead to improvements in sleep duration, sleep quality, or connection with 44 nature” and in text, page: 438-439: “ However, no significant changes were observed in sleep duration, sleep quality, or participants’ connection with nature following the intervention.” However, the sleep fragmentation index (table 1) increased in GE. And this result is not described in the results section.

Thank you for this opportunity

Author Response

Page 224-225: “For the sleep-related variables, a three-way ANOVA was performed, incorporating medication as a factor due to its potential influence on these outcomes.” This information appears only in statistical analysis without mentioning how these data were obtained

We thank the reviewer for this observation.

The use of pharmacological agents with potential effects on sleep duration and quality was assessed, including benzodiazepines, non-benzodiazepine hypnotics, sedative antidepressants, sedative antipsychotics, melatonin and melatonin receptor agonists, activating antidepressants, corticosteroids, beta-blockers, decongestants, and bronchodilators. Information regarding medication use was obtained during a medical evaluation conducted prior to participants’ enrollment in the study. This information was included in the methodology section.

Please see page 7, from line 196 to line 201

Page: 274-277: “Regarding sleep efficiency, the combined effect of intervention time (baseline vs. 16 weeks) and group (experimental vs. control) was a significant effect by the medication used by the participants (p=0.034), suggesting that the impact of the intervention varied according to the use of drugs with potential effects on sleep (partial η² = 0.062; observed power 0.571).” What drugs are the authors referencing?

Thank you for pointing this out.

This information is stated in the previous item.

The results in the tables are difficult to see due to the low resolution. This must be improved.

We appreciate the suggestion and we agree with the reviewer’s assessment. The resolution of the tables and figures was enhanced to ensure better visual quality.

Page 5 – Figure 1; Page 9 – Figure 2; Page 10 – Figure 3; Page 13 – Table 1; Page 14 – Table 2; Page 16 – Table 3

The sleep fragmentation index (table 1) increased in GE. And this result is not described in the results section. This section must be improved.

We thank the reviewer for this observation. We have sought to improve this section by introducing the following text:

“Both groups showed a reduction in TST (-7.23% in the EG and -11.86% in the CG) and SE (-3.56% and -2.59%, respectively), as well as an increase in the SFI (+25.45% and +7.39%, respectively). However, statistically significant differences (p≤ 0.05) were observed only in SFI (Table 1)”

Please see page 14, from line 301 to line 303

In pages 257-259 “After 16 weeks of intervention, 20% of the participants in the EG and 10% in the CG showed signs suggestive of possible sleep disorders (Table 2).” Are these results statistically different?

We appreciate the opportunity to explain this section of the results.

No statistical comparison of the observed percentages reported in the table was initially conducted in the study. However, although 20% of participants in the experimental group and 10% in the control group showed signs suggestive of possible sleep disturbances after 16 weeks of intervention, this difference was not statistically significant (Fisher’s exact test, p = 0.493). Data not expressed in the body of the text.

The author did not answer the study objective. The authors affirmed in abstract page 43-45: “however, it did not lead to improvements in sleep duration, sleep quality, or connection with 44 nature” and in text, page: 438-439: “However, no significant changes were observed in sleep duration, sleep quality, or participants’ connection with nature following the intervention.” However, the sleep fragmentation index (table 1) increased in GE. And this result is not described in the results section.

We thank the reviewer for this valuable observation. In response, we have revised the Results and Conclusions sections to read as follows

Results:

“However, when comparing the rates of change in these variables between the two groups, statistically significant differences (p ≤ 0.05) were observed solely for the SFI (Table 1).”

This revision explicitly clarifies that the SFI was the only variable to demonstrate statistical significance, thereby eliminating any potential ambiguity and ensuring consistency with the data presented in Table 1. We believe this change enhances the clarity and accuracy of the Results section and fully addresses the reviewer’s concern.

Please see page 14, from line 301 to line 303

Conclusions:

“However, the 16-week intervention did not result in significant changes in participants’ connection with nature, nor in sleep duration or efficiency. Nonetheless, an in-crease in the number of awakenings and transitions between sleep stages was observed in the EG compared to the CG, suggesting more fragmented and potentially less restorative sleep”

We believe that the revised paragraph offers greater clarity by explicitly distinguishing between the variables that did not change significantly (namely sleep duration, and sleep efficiency) and those that showed a notable difference (sleep fragmentation index).

Please see page 21, from line 489 to line 493

The presentation of the results should be improved - We appreciate the valuable suggestion for improving this section of the manuscript. We have revised the text accordingly, aiming to address the reviewer’s comments thoroughly and precisely. We hope that the revised version meets expectations and contributes to a clearer understanding of the presented results.

The conclusions should be better supported by the results - We thank the reviewer for this important observation. In response, we have carefully reviewed and revised the Conclusions section to ensure it is more clearly supported by the results presented in the manuscript.

All figures and tables must be clear and well-presented - We sincerely thank the reviewer for the valuable feedback provided. In response, we have thoroughly revised all figures and tables in the manuscript to improve their clarity, presentation, and overall quality.

The description of the methods can be improved - We thank the reviewer for the constructive suggestion. In response, we have revised the Methods section to improve its clarity and provide a more detailed and structured description of the procedures used. We hope that these adjustments meet the reviewer’s expectations and contribute to a better understanding of the methodological approach adopted in the study.

Round 2

Reviewer 2 Report

Comments and Suggestions for Authors

I appreciate the opportunity to review this manuscript.

The authors improved the manuscript regarding the reviewers' suggestions.

Best regards.